# pFedBBN: A Personalized Federated Test-Time Adaptation with Balanced Batch Normalization for Class-Imbalanced Data

## Abstract

Test-time adaptation (TTA) in federated learning (FL) is crucial for handling unseen data distributions across clients, particularly when faced with domain shifts and skewed class distributions. Class Imbalance (CI) remains a fundamental challenge in FL, where rare but critical classes are often severely underrepresented in individual client datasets. Although prior work has addressed CI during training through reliable aggregation and local class distribution alignment, these methods typically rely on access to labeled data or coordination among clients, and none address class unsupervised adaptation to dynamic domains or distribution shifts at inference time under federated CI constraints. Revealing the failure of state-of-the-art TTA in federated client adaptation in CI scenario, we propose **pFedBBN**, a personalized federated test-time adaptation framework that employs balanced batch normalization (BBN) during local client adaptation to mitigate prediction bias by treating all classes equally, while also enabling client collaboration guided by BBN similarity, ensuring that clients with similar balanced representations reinforce each other and that adaptation remains aligned with domain-specific characteristics. pFedBBN supports fully unsupervised local adaptation and introduces a class-aware model aggregation strategy that enables personalized inference without compromising privacy. It addresses both distribution shifts and class imbalance through balanced feature normalization and domain-aware collaboration, without requiring any labeled or raw data from clients. Extensive experiments across diverse baselines show that pFedBBN consistently enhances robustness and minority-class performance over state-of-the-art FL and TTA methods.

## 1 Introduction

Federated Learning (FL) enables decentralized training across a network of clients, such as smartphones, hospitals, or IoT devices, without sharing raw data. This is critical in privacy-sensitive domains like mobile computing, healthcare, and smart environments McMahan et al. (2017); Chen et al. (2025); Noble et al. (2022); Liu et al. (2024). However, data in FL is often non-identically distributed (non-IID), evolves over time, and suffers from issues such as client drift, system heterogeneity, and catastrophic forgetting, which significantly hinder model convergence and generalization Kairouz et al. (2021); Zhao et al. (2018). In such dynamic settings, Test-Time Adaptation (TTA) emerges as a crucial paradigm, enabling models to adapt on-the-fly to unseen distributions using only unlabeled test data. This adaptability is particularly important in federated environments where data distributions shift continuously across clients. However, relying solely on unlabeled data during adaptation can amplify prediction errors and even trigger catastrophic forgetting Niu et al. (2022), making the design of effective federated TTA both necessary and highly non-trivial.

Beyond distribution shifts, another fundamental challenge in FL is class imbalance (CI) Xiao & Wang (2021); Wang et al. (2021b); Seol & Kim (2023), where skew manifests both locally within individual client datasets and globally across the federation, systematically biasing model updates toward head classes while severely degrading tail-class performance Zhang et al. (2023). Existing CI mitigation techniques, such as resampling Khushi et al. (2021), data augmentation Duan et al. (2020), or cost-sensitive losses Sarkar et al. (2020); Khan et al. (2017), are ill-suited for FL. These methods assume access to raw data, which violates privacy, or fail when applied locally without co-

ordination. FL-specific solutions often require proxy servers Huang et al. (2016), auxiliary models Wang et al. (2017), or data exchange, introducing overhead and compromising privacy. Then, Wang et al. (2021b) proposed a federated learning approach to mitigate CI by adjusting training dynamics using labeled data across clients. Then, BalanceFL Shuai et al. (2022) employs a novel local update scheme that rectifies class imbalance by forcing each client's model to simulate training on a uniformly distributed dataset, thereby improving performance on underrepresented classes without violating privacy constraints. However, their FL method is designed for the supervised training phase where ground truth is available.

In Federated Test Time Adaptation (FTTA), the challenges compound. Without labels, correcting for both domain shifts and CI becomes significantly harder. Batch Normalization (BN) statistics, commonly used in TTA, become biased You et al. (2021) due to dominant classes, resulting in degraded adaptation. For instance, ATP Bao et al. (2023) reduces forgetting by learning per-module adaptation rates, but it assumes relatively stable test distributions. In contrast, FedICON Tan et al. (2023) improves representations through inter-client contrastive invariance, but its unsupervised refinement is still prone to being dominated by head classes. Similarly, FedTHE / FedTHE+ Jiang & Lin (2022) aim to balance global and personalized heads, yet their ensembling strategy often dilutes signals for underrepresented classes when clients face highly heterogeneous out-of-distribution data. Finally, FedTSA Zhang et al. (2024) enables similarity-guided collaboration using temporal–spatial correlations, but this approach relies on shared feature statistics that raise privacy and reconstruction risks. Recently, FedCTTA Rajib et al. (2025) addresses FTTA through a collaborative continual adaptation strategy through similarity-aware aggregation based on model output distributions of different clients. However, none of the existing approaches explicitly handle unlabeled TTA under both domain shifts and severe CI, which motivates the need for a new solution that is privacy-preserving, scalable, and robust to skewed class priors.

Moreover, Conventional batch normalization (BN) suffers from notable limitations when applied in test-time adaptation (TTA) under distribution shifts. Its frozen statistics at inference fail to generalize under distribution shifts Yuan et al. (2023); Nado et al. (2020); Gong et al. (2022); Lim et al. (2023); Niu et al. (2023). Generally, it aggregates statistics across all classes, leading to biased estimates dominated by majority classes. This results in internal covariate shift, misclassification of minority classes, and reduced macro-average accuracy in the existing FTTA methods.

To address these issues, we propose **pFedBBN**, the first framework that tackles Class Imbalance in Federated Test Time Adaptation. Our approach enables each client to adapt its model online to local distribution, unlabeled data while simultaneously benefiting from domain-similar peers through personalized aggregation. At the client side, we employ Class-Wise Adaptive Normalization (CWAN) with confidence-guided knowledge distillation to adapt on unlabeled data. Specifically, a balanced batch normalization (BBN) module replaced conventional BN to track per-class feature statistics using pseudo-labels and fuses them into balanced global estimates, mitigating majority-class bias and ensuring fair normalization. To further stabilize unsupervised updates, a confidence-filtered self-distillation approach selectively updates the model using pseudo-labeled data of teacher model. Only the BBN affine parameters are updated to preserve generalization. Once local adaptation is complete, we introduce a personalized cross-client collaboration step, where clients exchange only statistical descriptors and aggregate models based on domain similarity. Specifically, we exploit balanced batch normalization statistics as privacy-preserving domain descriptors to measure inter-client similarity. This facilitates a distance-aware, personalized aggregation strategy, where each client selectively integrates knowledge from peers with related distributions, yielding robust and domain-adaptive personalized models, all under strict privacy constraints.

In summary, the key contributions of this work are:

- We propose **pFedBBN**, the first framework specifically designed to address *class imbalance* in federated test-time adaptation, where clients adapt models using unlabeled, locally available test data under domain and class distribution shifts.

- We introduce a **Class-Wise Adaptive Normalization (CWAN)** module that maintains *per-class feature statistics* using pseudo-labels. By interpolating these with batch-level statistics, CWAN mitigates the bias introduced by dominant classes during adaptation without accessing ground-truth labels.

- To reduce the impact of noisy pseudo-labels, we selectively updates the model using only high-confidence pseudo-labeled samples via a teacher student based knowledge distillation, effectively minimizing error accumulation and catastrophic forgetting.

- We conduct comprehensive experiments on non-IID, class-imbalanced, and domain-shifted settings, demonstrating the effectiveness of pFedBBN over state-of-the-art FL and TTA baselines.

## 2 RELATED WORK

**Federated Learning (FL) and Test-Time Adaptation (TTA):** FL McMahan et al. (2017) enables collaborative model training across decentralized clients without directly sharing raw data, which is essential in privacy-sensitive domains such as healthcare, mobile computing, and IoT. However, challenges such as non-IID data distributions, class imbalance, and domain heterogeneity remain fundamental obstacles. Several works have explored methods to improve robustness under such settings, including communication-efficient optimization Chen et al. (2025), privacy-preserving learning with differential privacy Noble et al. (2022), and adaptive personalization Liu et al. (2024).

TTA methods aim to improve generalization under distribution shifts by adapting models using unlabeled test data. Recent work has shown the effectiveness of updating Batch Normalization (BN) statistics for adaptation You et al. (2021), although such methods often suffer from error accumulation and catastrophic forgetting in the absence of ground-truth supervision Niu et al. (2022). The BN statistics are particularly vulnerable to skewed class distributions, leading to biased adaptation under imbalance, hence degrading the performance of minority classes.

**Class Imbalance (CI) in FL:** CI is a critical challenge in FL, as dominant classes can overshadow rare yet important ones. Traditional solutions include resampling Khushi et al. (2021), data augmentation Duan et al. (2020), and cost-sensitive losses Sarkar et al. (2020); Khan et al. (2017). However, these methods typically assume access to centralized raw data, making them unsuitable for federated settings. FL-specific approaches have been proposed, including the use of proxy servers Huang et al. (2016), auxiliary models Wang et al. (2017), or data sharing among clients. Yet, such methods raise privacy and communication concerns. More recently, Wang et al. Wang et al. (2021b) proposed a federated optimization method that adjusts training dynamics across clients to address imbalance. However, this work focuses on the training phase and does not extend to test-time adaptation.

**Federated Test-Time Adaptation (FTTA):** FTTA is even more challenging, as clients face heterogeneous domain shifts, non-IID distributions, and the absence of labels. Recent TTA-in-FL methods use complementary strategies: ATP Bao et al. (2023) learns per-module adaptation rates during training and uses those learned rates to selectively adapt modules at test time via unsupervised entropy minimization. FedICON Tan et al. (2023) enforces inter-client invariance through contrastive representation learning during federation and performs unsupervised contrastive refinement at test time. FedTHE or FedTHE+ Jiang & Lin (2022) adopt a two-head design that combines a global generic classifier with a local personalized classifier to balance generalization and personalization at inference. And FedTSA Zhang et al. (2024) enables collaborative test-time adaptation by computing temporal–spatial correlations from local feature statistics to guide similarity-based aggregation using a memory bank and server-side aggregation. The recently proposed FedCTTA Rajib et al. (2025) introduced a collaborative continual adaptation strategy that improves robustness under distribution shifts. However, it does not address the challenge of class imbalance, which remains an open problem in FTTA.

## 3 METHODOLOGY

In real-world FL, clients face non-stationary, heterogeneous data that diverges from the source domain data. To ensure robust, personalized inference, we propose a unified framework that enables local test-time adaptation on unlabeled data and collaboration with similar domain clients measured via distance-aware aggregation. Our FL method includes two steps: unsupervised local adaptation of clients, and personalized aggregation among clients based on distribution similarity.

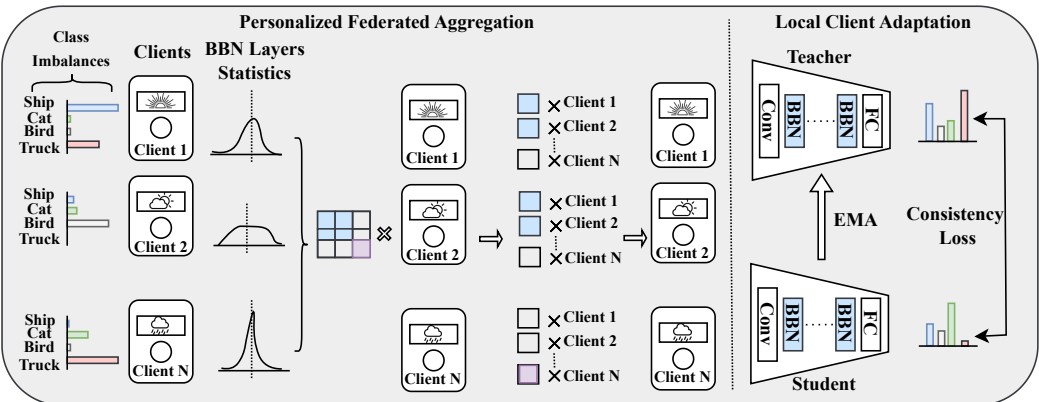

Figure 1: The overall framework of pFedBBN where each client performs unsupervised local adaptation using class-wise balanced batch normalization (BBN) and confidence-filtered distillation. Adapted batch normalization statistics are then used to compute client similarities, enabling personalized aggregation without sharing raw data.

### 3.1 UNSUPERVISED LOCAL CLIENT ADAPTATION

In the absence of labeled data, each client must adapt to its own distribution shift using only the pretrained source model. Our client-side test-time adaptation framework comprises two core strategies: Local Client Adaptation via Knowledge Distillation and Class-Wise Adaptive Normalization (CWAN) to address distributional drift to guide learning through self-supervision. To adapt each client model to its local domain while preserving generalization, we employ a lightweight self-training scheme with knowledge distillation in a teacher–student framework. We define a *teacher* network $f_t(x; \Theta_t)$ and a *student* network $f_s(x; \Theta_s)$, initialized from the pre-trained source model $f_{src}(x; \Theta_{src})$ and updated during test time. Prior to adaptation, the student model is equipped with CWAN by replacing standard batch normalization layers with class-aware normalization layers, enabling balanced statistics across classes during inference.

### 3.1.1 CLASS-WISE ADAPTIVE NORMALIZATION (CWAN)

We introduce CWAN, where conventional BN is replaced with Balanced BN that maintains separate statistics for each class, updated dynamically from pseudo-labels, and fuses them into balanced global estimates. This prevents majority-class bias and ensures fair normalization across categories. Since ground-truth labels are unavailable during TTA, CWAN leverages pseudo-labels from the teacher model to perform class-wise normalization.

Let $\mathbf{z}_i \in \mathbb{R}^d$ be the feature vector of test input $\mathbf{x}_i$, with pseudo-label $\hat{c}_i \in \{1, \ldots, K\}$. For each class $k$, CWAN keeps running estimates of mean $\mu_k^{(t)}$ and variance $\sigma_k^{2,(t)}$ at iteration $t$, which are updated as:

$$\mu_k^{(t)} = \mu_k^{(t-1)} + \Delta\mu_k^{(t)} \tag{1}$$

$$\sigma_k^{2,(t)} = \sigma_k^{2,(t-1)} - \left(\Delta\mu_k^{(t)}\right)^2 + \eta \sum_{i=1}^{B} \frac{\mathbf{1}(\hat{c}_i = k)}{d}\left[\left(\mathbf{z}_i - \mu_k^{(t-1)}\right)^2 - \sigma_k^{2,(t-1)}\right] \tag{2}$$

where $B$ is the batch size, $d$ is the feature dimension, $\eta$ is a momentum factor, and

$$\Delta\mu_k^{(t)} = \eta \sum_{i=1}^{B} \frac{\mathbf{1}(\hat{c}_i = k)}{d}\left(\mathbf{z}_i - \mu_k^{(t-1)}\right). \tag{3}$$

To avoid bias toward overrepresented classes, CWAN computes balanced global statistics by averaging across all $K$ classes:

$$\mu^{(t)} = \frac{1}{K}\sum_{k=1}^{K}\mu_k^{(t)}, \quad \sigma^{2,(t)} = \frac{1}{K}\sum_{k=1}^{K}\left(\sigma_k^{2,(t)} + (\mu_k^{(t)} - \mu^{(t)})^2\right). \tag{4}$$

### 3.1.2 LOCAL CLIENT ADAPTATION VIA KNOWLEDGE DISTILLATION

During local adaptation, each client leverages unlabeled test samples to update its student model. For an input $\mathbf{x}$ and its augmented counterpart $\tilde{\mathbf{x}}$, the probability outputs are:

$$\mathbf{p}^{(S)} = \text{softmax}(f_s(\mathbf{x}; \Theta_s)), \quad \mathbf{p}^{(T)} = \text{softmax}(f_t(\tilde{\mathbf{x}}; \Theta_t)), \quad \mathbf{p}^{(Src)} = \text{softmax}(f_{src}(\mathbf{x}; \Theta_{src})).$$

A reliability check is applied using entropy threshold to ignore highly uncertain samples. For such cases, we derive the pseudo-label from the teacher model, $f_t$, as $\hat{y} = \arg\max \mathbf{p}^{(T)}$. and update the student using a distillation loss:

$$\mathcal{L}_{\text{KD}} = \frac{1}{B} \sum_{i=1}^{B} \mathbb{I}\Big[H(\mathbf{p}_i^{(T)}) < \delta\Big] \cdot \text{CE}\Big(\hat{y}_i, \mathbf{p}_i^{(S)}\Big), \tag{5}$$

where $\delta$ is a confidence threshold, CE denotes cross-entropy and $B$ is the batch size.

To avoid overfitting and preserve transferability, adaptation is restricted to the affine parameters of the normalization layers in $f_s$, while the rest of the network remains unchanged.

In addition, to keep the adapting teacher close to the original source behavior, we add a *Consistency Regularization Loss*:

$$\mathcal{L}_{\text{CR}} = \frac{1}{BK_c} \sum_{i=1}^{B} \big\|\mathbf{p}_i^{(T)} - \mathbf{p}_i^{(Src)}\big\|_2^2, \tag{6}$$

where $\mathbf{p}_i^{(Src)}$ is the probability output of the fixed source model. This regularization anchors the adapting teacher to the pretrained source model, preventing catastrophic drift while still permitting gradual domain-specific refinement.

## 3.2 PERSONALIZED FEDERATED AGGREGATION

Local adaptation alone cannot guarantee robustness when client domains differ substantially. To avoid negative transfer while still leveraging cross-client knowledge, we design a similarity-aware aggregation mechanism that uses balanced batch-normalization (BN) statistics as a data-driven proxy for domain relatedness.

### 3.2.1 BATCH NORMALIZATION STATISTICS FOR DOMAIN SIMILARITY

Once local adaptation is complete, each client possesses a refined student model whose balanced batch normalization layers encode a statistical summary of the class-balanced local feature distribution during CI. These statistics form the basis for domain similarity estimation among clients used during aggregation.

Let $\mathcal{L}$ denote the set of BN layers, and for each client $i$ and layer $\ell \in \mathcal{L}$, let $\boldsymbol{\mu}_\ell^{(i)}, \boldsymbol{\sigma}_\ell^{2(i)} \in \mathbb{R}^d$ be the flattened global mean and variance vectors, respectively. The pairwise distance between clients $i$ and $j$ is given by:

$$D_{ij} = \frac{1}{|\mathcal{L}|} \sum_{\ell \in \mathcal{L}} \frac{1}{2} \left( \|\boldsymbol{\mu}_\ell^{(i)} - \boldsymbol{\mu}_\ell^{(j)}\|_2 + \|\boldsymbol{\sigma}_\ell^{2(i)} - \boldsymbol{\sigma}_\ell^{2(j)}\|_2 \right) \tag{7}$$

This similarity aggregation choice is analytical, BN statistics are sufficient to describe the latent feature distribution seen during adaptation (Figure 5), making $D_{ij}$ a compact yet informative similarity metric that respects privacy (no raw data sharing).

### 3.2.2 SIMILARITY-WEIGHTED COLLABORATION MATRIX

We derive a row-stochastic similarity matrix $W \in \mathbb{R}^{N \times N}$ from $D$ using a temperature-controlled softmax:

$$W_{ij} = \begin{cases} \dfrac{e^{-D_{ij}/\tau}}{\sum_{k \neq i} e^{-D_{ik}/\tau}} \left(1 - \omega_i\right), & j \neq i, \\ \omega_i, & j = i, \end{cases} \tag{8}$$

where $\omega_i = \left[1 + \sum_{k \neq i} e^{-D_{ik}/\tau}\right]^{-1}$.

This ensures each client gives more weight to peers with similar BBN statistics while preserving a degree of self-reliance.

### 3.2.3 AGGREGATED PERSONALIZED MODEL

Each client receives a personalized model computed as:

$$\theta_{\text{agg}}^{(i)} = \sum_{j=1}^{N} W_{ij} \cdot \theta^{(j)} \tag{9}$$

This model is then used for continued inference or further local adaptation on client $i$'s stream. Unlike FedAvg, which enforces a single global model, this strategy yields client-specific aggregates that exploit collaboration when beneficial, remain robust under severe class imbalance, and require only lightweight exchange of BN statistics.

## 4 RESULT AND DISCUSSION

**Dataset.** We conduct the experiments on two corruption benchmark datasets: CIFAR-10-C and CIFAR-100-C. These datasets were created by applying 15 different image corruptions (e.g., Gaussian noise, blur, brightness) at five discrete severity levels from 1 to 5 on the test sets of the CIFAR-10 and CIFAR-100 datasets, respectively. In this work, we focus exclusively on the worst-case noise level: severity-5, thereby simulating maximal domain shift to stress-test the model's robustness.

**Implementation Details.** We conduct experiments under both IID and Non-IID settings. Non-IID scenarios are simulated using Dirichlet sampling with concentration parameter $\delta \in \{0.005, 0.01, 0.1\}$, where lower $\delta$ induces higher class imbalance. For test-time adaptation, we evaluate Tent Wang et al. (2021a), CoTTA Wang et al. (2022), RoTTA Yuan et al. (2023), and ROID Marsden et al. (2024). In federated settings, we compare FedAvg McMahan et al. (2017), FedAvg-M Cheng et al. (2024), FedProx Li et al. (2020), pFedGraph Ye et al. (2023), and FedAMP Huang et al. (2021), with 10 simulated clients. Robustness under distribution shifts is assessed on CIFAR-10-C and CIFAR-100-C with 15 corruption types at severity level 5, using WideResNet-28 and ResNeXt-29, respectively. The batch size for each client is chosen to be 200.

### 4.1 COMPARISON

Table 1 presents a comprehensive performance comparison across various federated learning aggregation strategies and test-time adaptation (TTA) methods, evaluated on two corruption-augmented benchmarks: CIFAR-10-C and CIFAR-100-C. The experiments are conducted in both IID and Non-IID client data settings, where Non-IID scenarios are simulated using Dirichlet distributions by varying the concentration parameter $\delta \in \{0.005, 0.01, 0.1\}$. Lower values of $\delta$ induce higher class imbalance across clients, thereby increasing the difficulty of the federated learning.

**Performance under IID Setting.** In the IID case, where data is evenly distributed across clients without class imbalance, most existing TTA methods such as Tent, CoTTA, and ROID demonstrate strong performance when paired with standard federated aggregation techniques like FedAvg and FedProx. For instance, Tent and CoTTA achieve over 81% accuracy on CIFAR-10-C across multiple aggregation strategies. Our proposed BBN method remains competitive, achieving consistent performance over 70% and 59% on the CIFAR-10-C and CIFAR-100-C datasets respectively.

**Performance under Non-IID Setting.** Under the more realistic Non-IID setting with varying degrees of class imbalance (controlled by Dirichlet $\delta$), the performance of standard TTA methods degrades significantly. In particular, methods like Tent and CoTTA drop to as low as 6–30% accuracy

Table 1: Comparison of federated learning (FL) methods with different test-time adaptation (TTA) techniques on CIFAR-10-C and CIFAR-100-C benchmark datasets with 10 clients. Results (Global Accuracies) are reported for both IID and non-IID settings (Dirichlet partitioning with $\delta \in 0.005, 0.01, 0.1$). Gray-highlighted rows correspond to our proposed BBN module, while the last row (pFedBBN) represents our full framework.

| Fed Method | TTA Method | CIFAR-10-C | | | | CIFAR-100-C | | | |
| | | IID | Non-IID | | | IID | Non-IID | | |
| | | | $\delta$=0.005 | $\delta$=0.01 | $\delta$=0.1 | | $\delta$=0.005 | $\delta$=0.01 | $\delta$=0.1 |
|---|---|---|---|---|---|---|---|---|---|
| Any | Source | 56.51 | 58.28 | 58.58 | 56.11 | 54.28 | 54.58 | 57.14 | 53.03 |
| None | Tent | 81.08 | 23.99 | 24.98 | 31.23 | 68.59 | 6.84 | 9.41 | 28.19 |
| | CoTTA | 81.78 | 21.13 | 23.42 | 31.06 | 66.24 | 6.01 | 10.88 | 28.94 |
| | ROID | 81.44 | 14.60 | 15.95 | 26.20 | 69.41 | 7.46 | 11.84 | 38.38 |
| | RoTTA | 66.63 | 30.64 | 32.73 | 51.06 | 50.60 | 15.03 | 28.31 | 44.61 |
| | BBN | 72.32 | 56.88 | 52.33 | 70.53 | 59.32 | 64.07 | 72.31 | 66.21 |
| FedAvg | Tent | 81.19 | 21.51 | 22.47 | 29.42 | 68.13 | 5.06 | 7.82 | 25.27 |
| | CoTTA | 81.61 | 19.92 | 21.11 | 28.74 | 65.94 | 6.08 | 9.32 | 32.53 |
| | ROID | 81.85 | 22.93 | 23.96 | 32.15 | 69.06 | 6.85 | 10.97 | 37.01 |
| | RoTTA | 64.16 | 64.54 | 64.34 | 64.68 | 49.15 | 49.46 | 55.00 | 49.07 |
| | BBN | 72.61 | 68.47 | 69.04 | 74.05 | 60.08 | 67.49 | 71.30 | 63.03 |
| FedProx | Tent | 81.20 | 21.52 | 22.47 | 29.42 | 68.13 | 5.06 | 7.82 | 25.27 |
| | CoTTA | 81.59 | 19.92 | 23.69 | 30.98 | 65.95 | 6.07 | 10.85 | 29.19 |
| | ROID | 81.81 | 20.87 | 24.02 | 31.77 | 69.30 | 6.86 | 10.69 | 37.01 |
| | RoTTA | 67.31 | 39.39 | 64.41 | 64.69 | 49.82 | 22.66 | 55.04 | 48.98 |
| FedAvgM | Tent | 81.30 | 21.45 | 22.37 | 29.38 | 67.52 | 5.05 | 7.82 | 25.39 |
| | CoTTA | 81.95 | 19.88 | 21.45 | 28.50 | 66.12 | 6.42 | 9.01 | 32.75 |
| | ROID | 81.87 | 23.07 | 24.09 | 32.35 | 68.81 | 6.87 | 11.11 | 37.04 |
| | RoTTA | 38.88 | 39.73 | 39.42 | 37.86 | 17.95 | 34.36 | 38.54 | 21.56 |
| pfedGraph | Tent | 81.19 | 21.52 | 22.49 | 29.98 | 68.11 | 5.12 | 7.83 | 25.04 |
| | CoTTA | 81.64 | 19.94 | 22.34 | 29.01 | 65.99 | 6.08 | 10.88 | 29.03 |
| | ROID | 81.84 | 22.99 | 23.96 | 32.34 | 69.07 | 5.88 | 10.96 | 36.91 |
| | RoTTA | 64.87 | 63.75 | 64.43 | 63.98 | 49.51 | 52.99 | 54.88 | 54.63 |
| | BBN | 72.73 | 69.37 | 66.89 | 73.54 | 60.14 | 67.29 | 72.61 | 63.71 |
| FedAmp | Tent | 81.19 | 20.65 | 22.49 | 28.28 | 68.15 | 5.09 | 7.85 | 25.30 |
| | CoTTA | 81.61 | 20.19 | 22.93 | 30.21 | 65.95 | 6.07 | 10.23 | 28.71 |
| | ROID | 81.83 | 21.11 | 23.82 | 33.11 | 69.11 | 6.88 | 11.21 | 36.88 |
| | RoTTA | 65.91 | 63.87 | 62.85 | 63.16 | 50.27 | 46.82 | 53.91 | 49.35 |
| | BBN | 72.36 | 63.16 | 61.36 | 72.53 | 59.52 | 69.88 | 72.62 | 63.29 |
| pFedBBN | BBN | 70.11 | 72.41 | 71.96 | 68.53 | 65.29 | 71.96 | 73.88 | 64.29 |

in extreme imbalance cases ($\delta = 0.005$) across both datasets. In contrast, our proposed TTA method BBN maintains robust performance across all levels of imbalance. For example, with FedAvg at $\delta = 0.005$, BBN achieves 68.47% on CIFAR-10-C and 67.49% on CIFAR-100-C, substantially outperforming other methods.

**Superior Performance of pFedBBN.** Notably, our full framework, pFedBBN, which combines personalized federated learning with the BBN-based TTA strategy, consistently achieves the highest or near-highest accuracy across all scenarios. It maintains stable performance even in highly imbalanced cases. For instance, on CIFAR-10-C with $\delta = 0.005$, pFedBBN achieves 72.41%, surpassing all other combinations of FL and TTA strategies. Similarly, on CIFAR-100-C, it reaches 73.88% accuracy at $\delta = 0.01$, demonstrating both robustness and effectiveness in diverse settings. These results show that while existing TTA methods falter under class imbalance and data heterogeneity, our BBN-based TTA, especially with pFedBBN, offers a more effective solution.

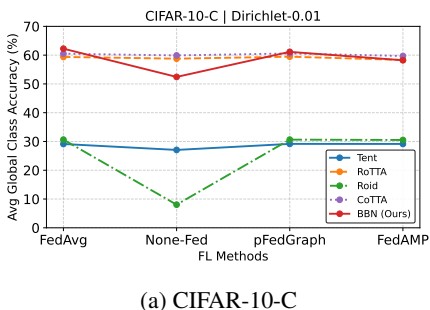 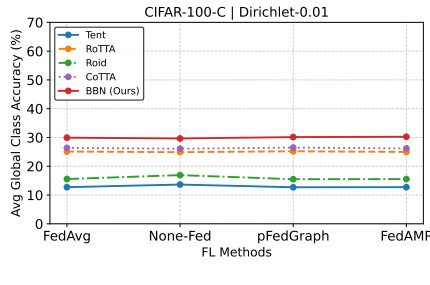

(a) CIFAR-10-C  (b) CIFAR-100-C

Figure 2: **Average global class accuracies (%) of different TTA methods under different Federated learning Schemes.** Our method (BBN) consistently delivers best performance under different federated setups for both CIFAR-10-C and CIFAR-100-C datasets.

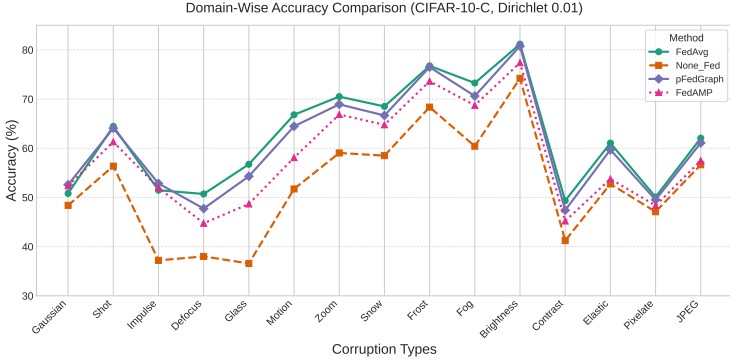

Figure 3: Domain-wise accuracy comparison of BBN under different Federated setups (Dirichlet $\delta = 0.01$) across CIFAR-10-C corruptions.

## 4.2 ANALYTICAL INSIGHTS

**Comparison of TTA setups under heterogeneity.** In Fig. 2a and Fig. 2b respectively, we show the performances on CIFAR-10-C and CIFAR-100-C datasets respectively for various test time adaptation techniques. For, CIFAR-10-C, provides almost consistent performance under all fed setups, while RoTTA falls slightly short. Our method is comparable to the techniques, and beats both the aforementioned techniques for pFedGraph aggregation. In Fig. 2b for the evaluation of CIFAR-100-C corruption dataset, our method consistently outperforms all the TTA methods under all aggregation setups. These results demonstrate BBN's robustness under various domain shifts.

**Performance analysis under domain shift** In Fig. 3 we demonstrate BBN's performance across different domains under class imbalance ($\delta$) = 0.01 for various aggregation techniques. We observe that for FedAvg aggregation our method performs the best across a diversity of corruption settings. This indicates better generalization under FedAvg aggregation. The personalized setup, pFedGraph comes in second with competitive performance for various corruptions eg: Gaussian Noise, Shot Noise, Fog, Brightness, JPEG compression, etc. From the figure, it is also evident that our method performs best in lighting and weather corruptions, modestly on noise corruptions, but struggles with blur, compression, and geometric distortions.

**Performance analysis of Major and Minor Classes for pFedBBN** For each client, the major class is defined as the class with the highest number of local samples, while the minor class is the one with the fewest samples. Fig. 4 shows the accuracy values for the major and minor classes on CIFAR-10-C dataset for each client under pFedBBN adaptation strategy. It can be observed from Fig. 4 that almost all the clients perform equally well for the major and minor classes under pFedBBN adaptation strategy under class imbalance ($\delta$ = 0.01). This shows the efficacy of pFedBBN in improving the performance of minor classes under the severe challenge of both data heterogeneity and varying domains across the clients.

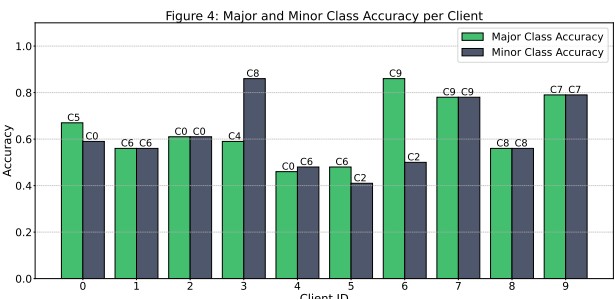

Figure 4: **Major and Minor Class accuracy per client for CIFAR-10-C (Dirichlet $\delta$ = 0.01) using pFedBBN strategy.** The ten classes of CIFAR-10-C are denoted by the symbols from C0 to C9. The accuracies for major and minor classes are similar for almost all the clients, demonstrating mitigation of the class-imbalance issue.

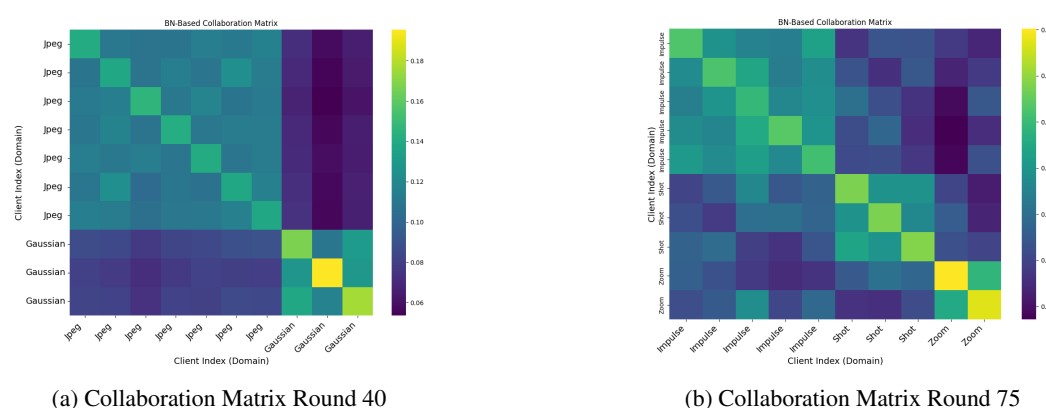

(a) Collaboration Matrix Round 40              (b) Collaboration Matrix Round 75

Figure 5: Collaboration Matrix (round = 40, 75) of our pFedBBN (total 75 federated rounds) which indicates which client give more priority while aggregating and it has clearly be seen that similar domains are aggregated more than others

**Collaboration Weight Analysis of Client Aggregation in pFedBBN** We compute a symmetric distance matrix based on the Balanced Batch Normalization (BBN) statistics, specifically, the global mean and variance, across all clients. This matrix is then used to derive the collaboration weights for aggregation. The results reveal that clients tend to prioritize aggregation with others from the same domain, highlighting domain-aware collaboration. As shown in Fig. 5, clients from similar domains consistently form clusters in the collaboration weight matrix across different federated rounds, indicating effective domain-wise distribution alignment.

## 5 CONCLUSION

We introduced pFedBBN, a federated test-time adaptation framework that addresses class imbalance and domain shifts by leveraging Balanced Batch Normalization. pFedBBN enables unsupervised, privacy-preserving client-side adaptation and introduces a class-aware server aggregation strategy. Experiments on benchmark datasets show that FedBBN improves robustness and minority-class performance over existing methods. This makes it a practical and scalable solution for real-world federated learning scenarios with non-IID and unlabeled test distributions. Moreover, our analysis demonstrates that pFedBBN not only balances major and minor class performance across heterogeneous clients but also adapts effectively under diverse corruption domains, confirming its robustness in challenging federated environments. These findings highlight its potential for deployment in safety-critical and privacy-sensitive applications such as healthcare, autonomous systems, and mobile platforms.

## REPRODUCIBILITY STATEMENT

All the results in this work are reproducible. We provide all the necessary code in the Supplementary Material to replicate our results. The repository includes environment configurations, scripts, and other relevant materials. We discuss the experimental settings in Section 4, including implementation details such as models, datasets, hyperparameters, etc.

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

## A    APPENDIX

## B    PERSONALIZED FEDERATED BALANCED BATCH NORMALIZATION

The pFedBBN algorithm is a personalized federated learning setup designed to tackle the data heterogeneity and class imbalance problem in test-time adaptation (TTA). This method focuses on Batch normalization statistics and a unique aggregation strategy. The process begins with the server broadcasting the current global model to all participating clients. Each client then independently performs a balanced batch normalization in test time adaptation on its local, unlabeled test data. This client-side adaptation is crucial as it not only fine-tunes the model to the client's specific data distribution but also addresses class imbalance by using a balanced loss function and pseudo label generated at test time as ground truth is not available. After local adaptation, each client extracts and sends its Batch Normalization (BN) statistics (mean and variance for each layer) back to the server. The server, instead of a simply averaging, uses the collected BN statistics to compute a collaboration matrix. This matrix quantifies the similarity between the data distributions of different clients, effectively utilizing the information of relatedness among the clients. A higher similarity between two clients' BN statistics results in a stronger collaboration weight. The server uses this collaboration matrix to perform a weighted aggregation, creating a personalized aggregated model for each client. This approach ensures that clients with similar data distributions contribute more to each other's model updates. This iterative process allows the global model to converge while maintaining the personalized characteristics of each client's model, making it robust to non-IID data.

## C    ADDITIONAL ANALYSIS

### C.1    PERFORMANCE GAIN OF PFEDBBN OVER BASELINES

We illustrate the performance gain of our proposed pFedBBN method over several standard and advanced federated learning baselines across Fig. 7 to Fig. 9, all of which are combined with our client-side BBN TTA. These figures provide a clear visual comparison of how our novel BN statistics aggregation strategy, a core component of pFedBBN, improves upon existing federated methods. Specifically, Fig. 7 compares pFedBBN against the FedAvg baseline, Fig. 7 highlights the improvements against the personalized graph-based pFedGraph, and Fig. 9 demonstrates the gain over the personalized FedAmp method. The consistent positive accuracy gains across various non-IID degrees on both CIFAR-10-C and CIFAR-100-C datasets serve as strong evidence that our aggregation approach is more effective at leveraging BN statistics for robust, personalized model updates.

### C.2    CLASS-WISE PERFORMANCE BALANCE UNDER DOMAIN SHIFT

Fig. 10 demonstrates the effectiveness of our method (pFedBBN) in mitigating prediction bias under domain shift. The paired accuracies for the Major Class and Minor Class very close across all 15 corruption domains of CIFAR-10-C.

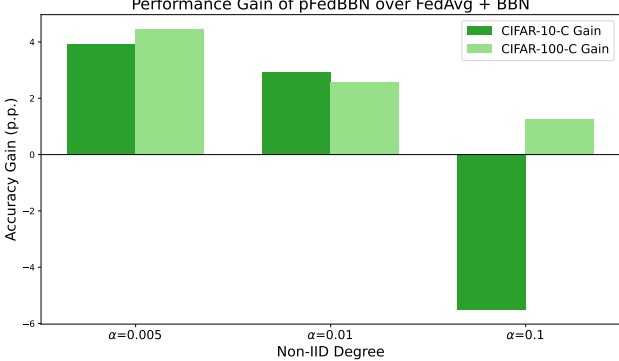

Figure 7: pFedBBN gain over FedAvgBBN.

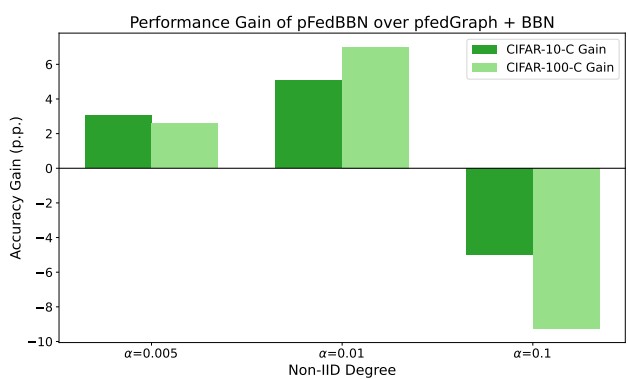

Figure 8: pFedBBN gain over pFedGraphBBN.

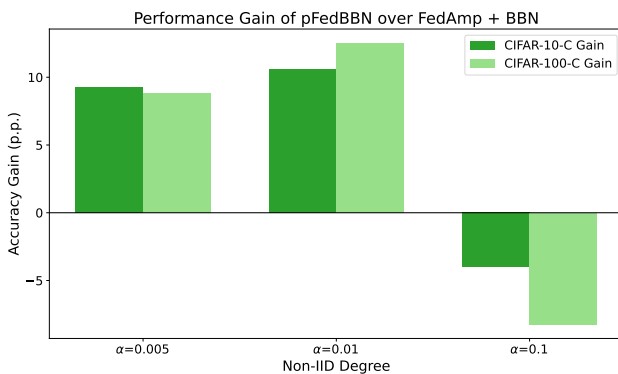

Figure 9: pFedBBN gain over FedAMPBBN.

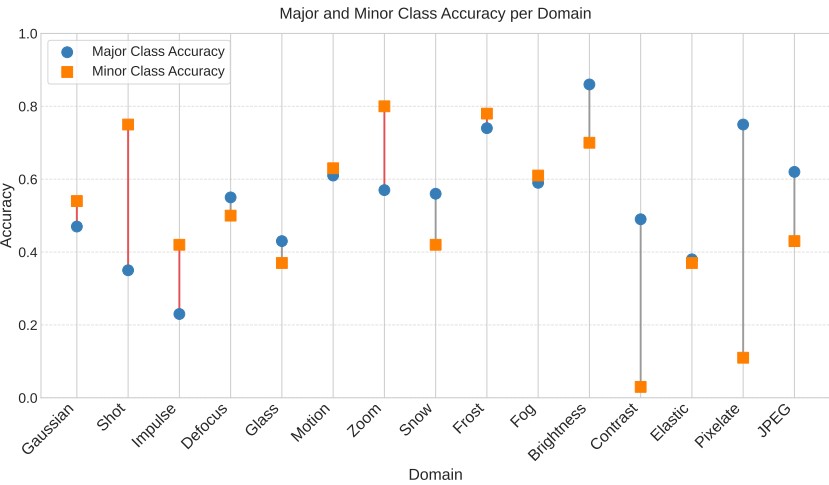

Figure 10: Major and minor class accuracy per domain for CIFAR-10-C dataset (Dirichlet $\delta = 0.01$) using pFedBBN strategy.

### C.3 CLASS IMBALANCE ANALYSIS

Fig. 11 and Fig. 12 provide a detailed analysis of the simulated class imbalance, which is crucial for understanding the challenges of the non-IID setting. Fig. 11, the Global Class Frequency plot, visually represents the severe class imbalance inherent in the CIFAR-10-C data distribution with a

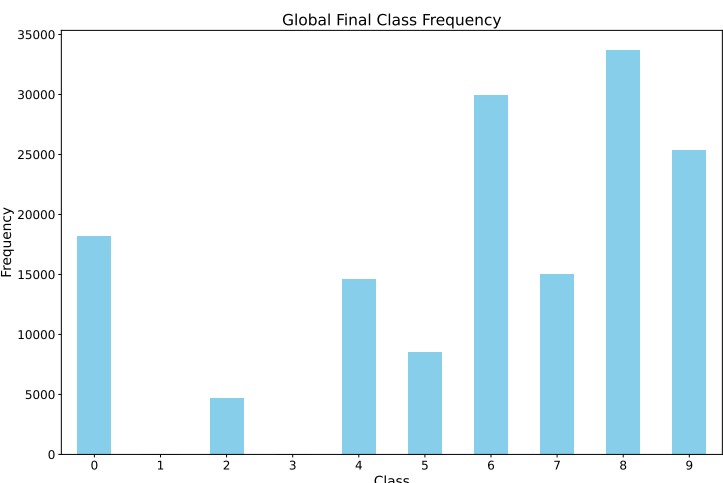

Figure 11: Global class frequency for CIFAR-10-C.

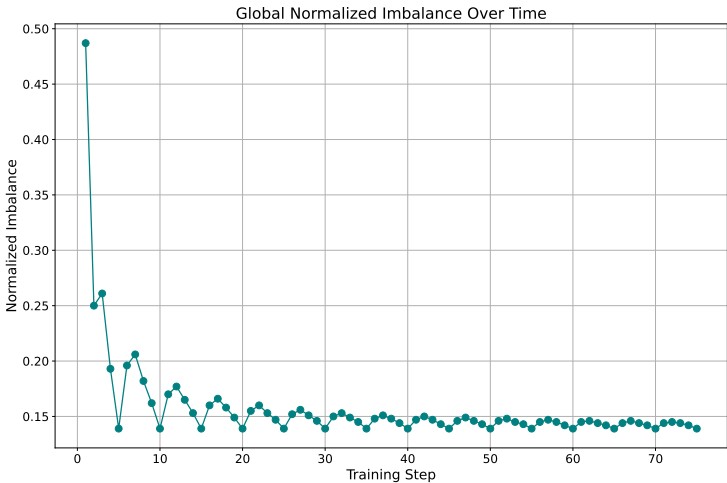

Figure 12: Global imbalance ratio with federated rounds.

Dirichlet $\delta$ of 0.005. It highlights the long-tail problem where a small number of classes dominate the dataset, while others are significantly under-represented. This imbalance is a primary driver of poor model performance in federated learning. Fig. 12, the Global Normalized Imbalance Over training steps plot, offers an insightful look into the training dynamics. This line graph tracks the normalized imbalance across training steps (federated rounds), showing how the model, despite the initial data disparity, learns to progressively correct for the imbalance. This demonstrates the effectiveness of the training process in mitigating dataset heterogeneity, a key objective of our method.

### C.4 PERFORMANCE COMPARISON OF FED TTA SETUPS

A performance comparison across federated methods and Test-Time Adaptation (TTA) strategies is presented across Fig. 13 to Fig. 18. A key observation across all scenarios is the distinct performance clusters formed by the TTA methods. Figures 13, 14, and 15 show that on the CIFAR-10-C dataset, TTA methods like Tent, CoTTA, and ROID consistently underperform, with accuracy scores generally below 30%. In stark contrast, RoTTA and BBN achieve significantly higher accuracies, often exceeding 60% and 70% respectively, making them the superior choices for TTA. This trend is echoed in the CIFAR-100-C results, as shown in Figures 16, 17, and 18, where Tent, CoTTA, and ROID again exhibit very low accuracy, while RoTTA and BBN maintain high performance

levels. Notably, the federated methods—FedAvg, pfedGraph, and FedAmp—show comparable performance with a given TTA method. However, the BBN TTA method paired with these federated methods generally yields the highest overall accuracy.

## C.5 IID ANALYSIS

In Figures 19 and 20, we provide a foundational analysis of model performance under IID (Independent and Identically Distributed) conditions, a crucial baseline for evaluating federated learning methods. The plots for CIFAR-10-C and CIFAR-100-C respectively, demonstrate that when data is evenly distributed across clients, all federated methods—both with and without TTA (Test-Time Adaptation) methods like BBN—achieve high and comparable accuracy. The lack of significant performance gaps in this setting confirms that the primary challenge of federated learning is not the distributed nature of the data itself, but rather the data heterogeneity introduced by non-IID distributions.

## C.6 DOMAIN-WISE PERFORMANCE FOR CIFAR-100-C

Fig. 21 shows the performance of our BBN TTA method on the complex CIFAR-100-C dataset (100 classes) under non-IID conditions ($\delta$=0.01) under various Federated aggregation techniques. The plot shows that pFedGraph maintains the highest overall accuracy across nearly all 15 corruption types. This consistent and significant outperformance confirms that the graph-based domain-aware collaboration mechanism is highly effective at preserving generalization capability and resilience in high-dimensional, severely imbalanced federated TTA environments, outperforming all tested baselines.

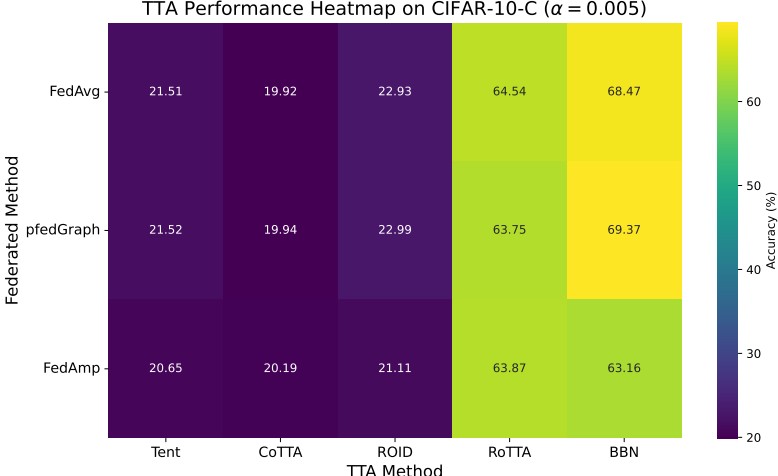

Figure 13: Fed TTA performance on CIFAR-10-C ($\delta = 0.005$)

## C.7 ROBUSTNESS ANALYSIS ON CIFAR-10-C & CIFAR-100-C FOR HETEROGENEITY ($\delta = 0.01$)

Table 2 shows the performances of various TTA methods combined with various federated learning setups under data heterogeneity ($\delta = 0.01$). From the table, it is evident that, our proposed TTA method (BBN) outperforms all the other TTA methods on average under all the Federated setups. For CIFAR-10-C BBN performs the best on FedAvg, while being competitive for pFedGraph and FedAMP. While for CIFAR-100-C the graph-based aggregation techniques perform slightly better than FedAvg.

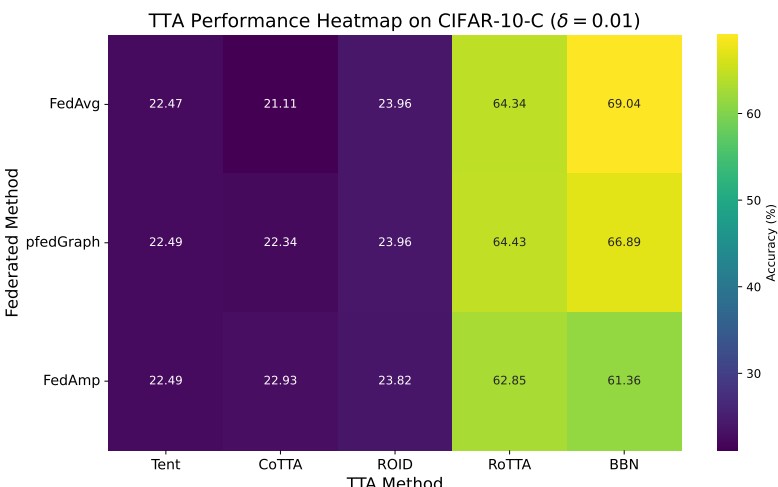

Figure 14: Fed TTA performance on CIFAR-10-C ($\delta = 0.01$)

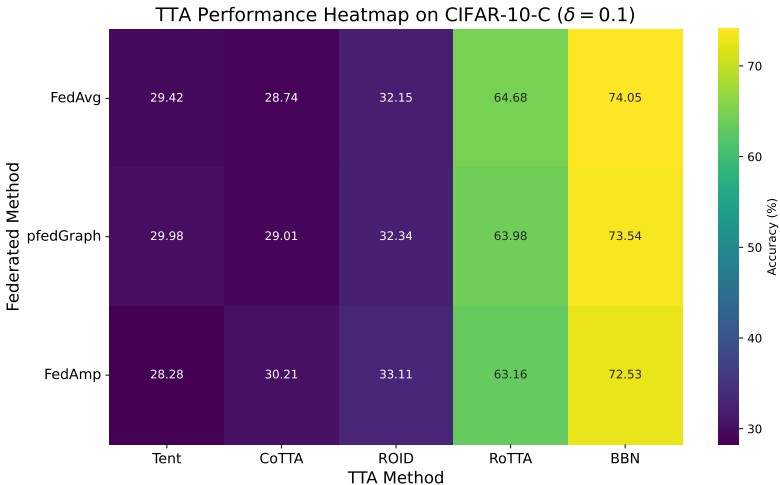

Figure 15: Fed TTA performance on CIFAR-10-C ($\delta = 0.1$)

## C.8 CLASS-WISE PERFORMANCE ANALYSIS FOR CIFAR-10-C ($\delta = 0.01$)

From Table 3, we observe the class-wise accuracies of the 10 classes in the CIFAR-10-C corruption dataset. Due to the simulated heterogeneity ($\delta = 0.01$), we see that no client receives samples from class-3 (C3). From the table, we can observe that BBN performs the best on average for all classes under various federated setups.

## C.9 DISTRIBUTION OF CLASS SAMPLES

Table 4 to Table 5 show the distribution of class samples across the various clients in each federated round for class-imbalance simulated using $\delta = 0.01$. For each federated round each client receives a batch of 200 samples. Therefore, the total count of samples being 2000 in each federated round as there are 100 clients in total. Due to simulating class imbalance via Dirichlet sampling ($\delta = 0.01$), the class-3 samples are not observed by any clients.

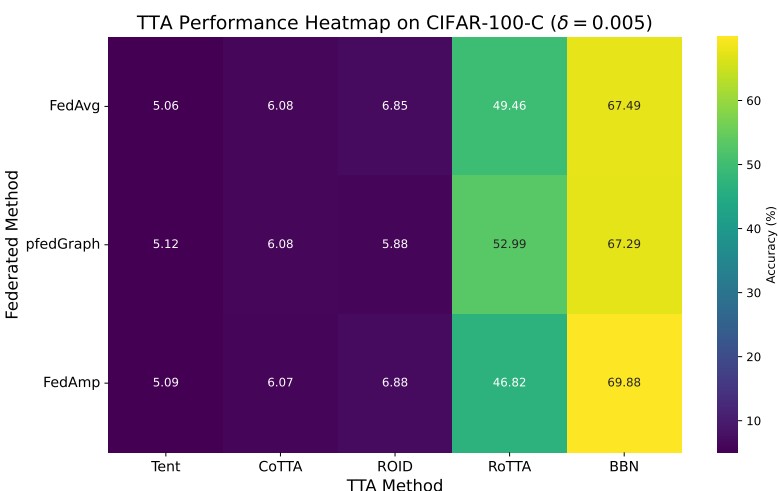

Figure 16: Fed TTA performance on CIFAR-100-C ($\delta = 0.005$)

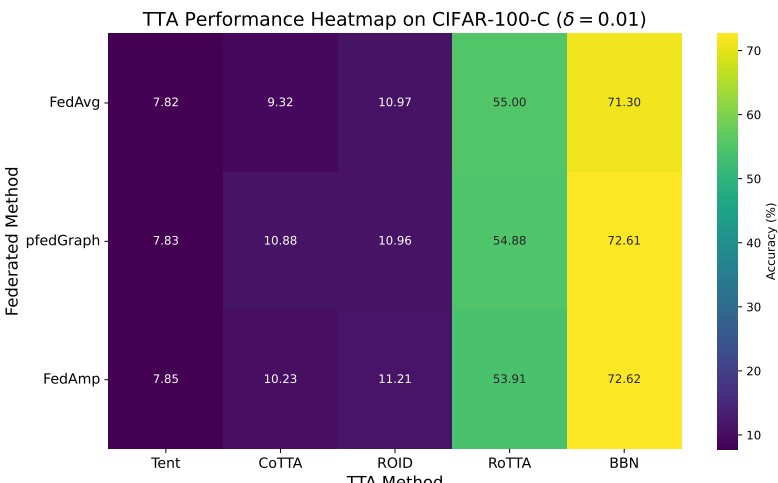

Figure 17: Fed TTA performance on CIFAR-100-C ($\delta = 0.01$)

## C.10 DOMAIN-WISE AVERAGE ACCURACY ON CIFAR-10-C ($\delta = 0.01$)

Table 6 shows the averages accuracies for all the clients for each domain for all the TTA setups under various federated aggregation strategies. On average, BBN outperforms rest of the TTA methods under all the federated aggregation techniques.

## C.11 CLIENT-WISE ACCURACY ON CIFAR-10-C ($\delta = 0.01$)

We show the client-wise performance across various TTA setups combined with different federated learning strategies in Table 7. On average we find that BBN outperforms all the other TTA techniques for all the clients under all the Federated aggregation schemes. We observe that BBN performs the best under the pFedGraph aggregation strategy.

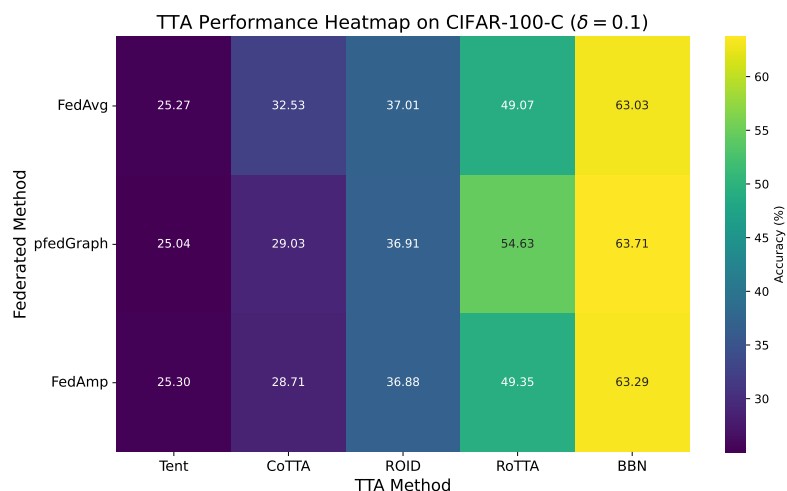

Figure 18: Fed TTA performance on CIFAR-100-C ($\delta = 0.1$)

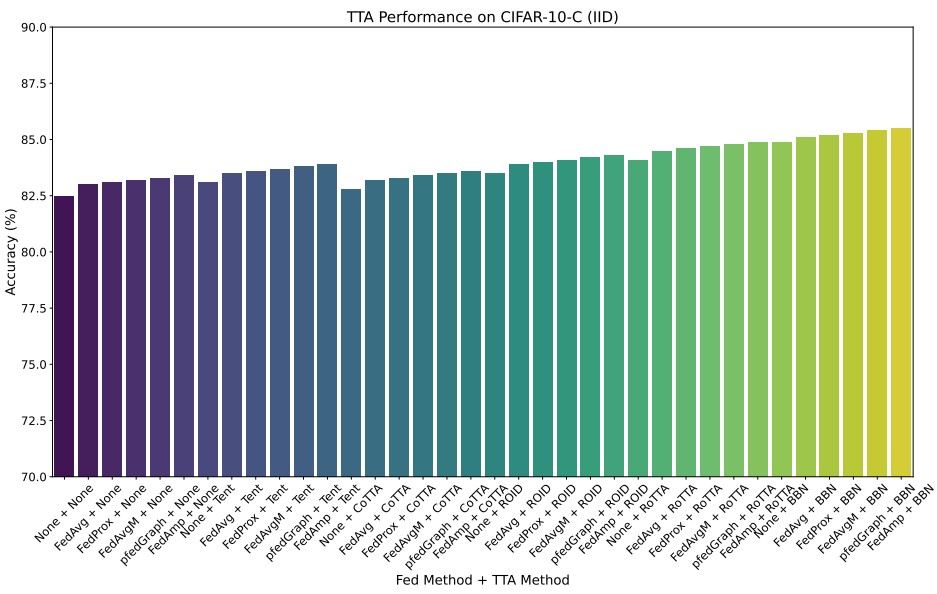

Figure 19: CIFAR-10-C IID performance.

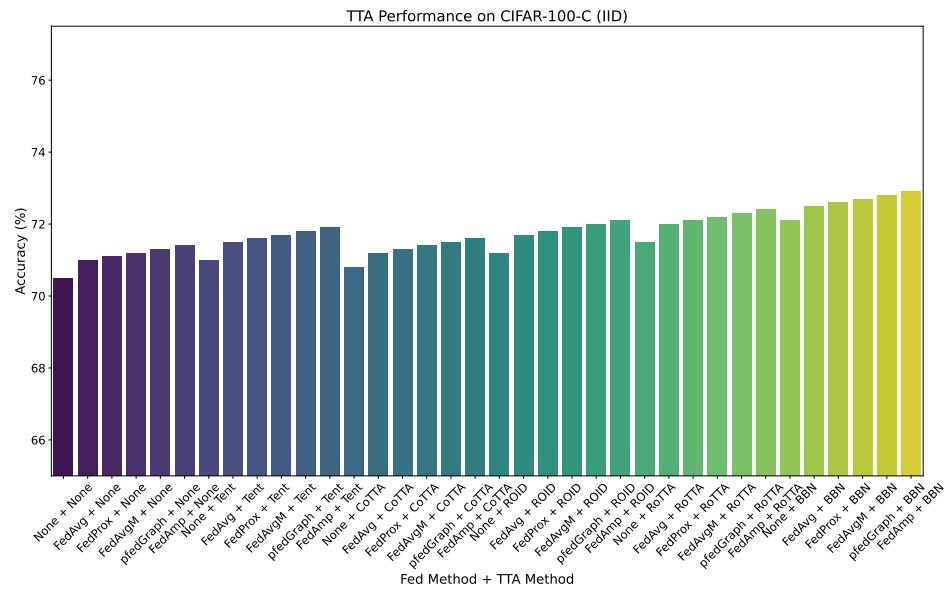

Figure 20: CIFAR-100-C IID performance.

Table 2: Average global class accuracy (%) on CIFAR-10-C and CIFAR-100-C under Dirichlet heterogeneity $\delta = 0.01$. Results are grouped for both CIFAR-10-C and CIFAR-100-C datasets across the various Federated TTA combinations. Higher values indicate better robustness to distribution skew.

| Dataset | TTA Method | Federated Algorithm | | | |
|---|---|---|---|---|---|
| | | None-Fed | FedAvg | pFedGraph | FedAmp |
| CIFAR-10-C | Tent | 23.90 | 29.12 | 29.14 | 29.05 |
| | CoTTA | 25.50 | 30.00 | 30.10 | 30.00 |
| | ROID | 22.00 | 30.64 | 30.64 | 30.52 |
| | RoTTA | 36.42 | 59.37 | 59.46 | 58.33 |
| | BBN | 52.43 | 62.24 | 61.14 | 58.22 |
| CIFAR-100-C | Tent | 10.30 | 12.73 | 12.71 | 13.04 |
| | CoTTA | 12.00 | 14.50 | 14.70 | 14.60 |
| | ROID | 12.00 | 15.56 | 15.49 | 15.54 |
| | RoTTA | 17.31 | 25.11 | 25.19 | 24.97 |
| | BBN | 29.65 | 29.92 | 30.13 | 30.25 |

Table 3: Class-wise global accuracy (%) on CIFAR-10-C (severity-5) under Dirichlet heterogeneity $\delta = 0.01$. We compare five TTA methods (Tent, CoTTA, ROID, RoTTA, BBN) across four federation schemes (None, FedAvg, pFedGraph, FedAMP).

| Method | Fed | C0 | C1 | C2 | C3 | C4 | C5 | C6 | C7 | C8 | C9 | Avg |
|---|---|---|---|---|---|---|---|---|---|---|---|---|
| Tent | None-Fed | 21.5 | 67.5 | 30.0 | 0.0 | 35.0 | 42.0 | 20.0 | 45.0 | 34.0 | 44.0 | 33.9 |
| | FedAvg | 28.0 | 82.0 | 50.0 | 0.0 | 56.0 | 62.0 | 65.0 | 74.0 | 62.0 | 80.0 | 55.9 |
| | pFedGraph | 28.3 | 81.8 | 49.5 | 0.0 | 56.5 | 61.8 | 61.5 | 73.0 | 60.5 | 79.2 | 55.3 |
| | FedAMP | 27.8 | 81.0 | 48.5 | 0.0 | 55.0 | 61.2 | 58.0 | 72.5 | 58.5 | 78.0 | 54.0 |
| CoTTA | None-Fed | 23.0 | 68.0 | 32.0 | 0.0 | 36.5 | 43.0 | 22.0 | 47.0 | 35.5 | 46.0 | 35.0 |
| | FedAvg | 29.5 | 83.0 | 52.0 | 0.0 | 58.0 | 63.5 | 66.5 | 75.5 | 63.5 | 81.5 | 57.3 |
| | pFedGraph | 29.7 | 82.9 | 51.5 | 0.0 | 58.4 | 63.1 | 63.1 | 74.2 | 62.0 | 80.6 | 56.6 |
| | FedAMP | 29.1 | 82.5 | 50.5 | 0.0 | 57.1 | 62.7 | 60.2 | 73.5 | 60.2 | 79.5 | 55.6 |
| ROID | None-Fed | 19.0 | 62.0 | 28.0 | 0.0 | 32.0 | 39.0 | 18.0 | 42.0 | 31.0 | 42.0 | 31.3 |
| | FedAvg | 30.6 | 81.4 | 54.0 | 0.0 | 59.0 | 65.0 | 67.0 | 76.5 | 65.0 | 83.0 | 58.2 |
| | pFedGraph | 30.5 | 81.3 | 53.2 | 0.0 | 59.2 | 64.6 | 64.0 | 75.0 | 63.2 | 81.9 | 57.3 |
| | FedAMP | 30.2 | 80.9 | 52.0 | 0.0 | 57.6 | 63.8 | 61.0 | 74.0 | 61.5 | 80.3 | 56.1 |
| RoTTA | None-Fed | 47.72 | 57.33 | 41.10 | 0.0 | 30.11 | 33.58 | 39.83 | 33.22 | 39.37 | 41.91 | 36.42 |
| | FedAvg | 69.21 | 86.67 | 54.09 | 0.0 | 58.96 | 63.56 | 66.00 | 66.31 | 60.89 | 67.97 | 59.37 |
| | pFedGraph | 69.76 | 85.33 | 53.25 | 0.0 | 59.25 | 65.89 | 63.94 | 67.29 | 61.73 | 68.13 | 59.46 |
| | FedAMP | 66.36 | 84.00 | 54.16 | 0.0 | 57.35 | 64.83 | 61.41 | 67.20 | 59.60 | 68.43 | 58.33 |
| BBN | None-Fed | 47.33 | 84.00 | 51.46 | 0.0 | 55.97 | 68.61 | 36.86 | 64.67 | 47.85 | 67.57 | 52.43 |
| | FedAvg | 63.48 | 85.33 | 50.96 | 0.0 | 57.20 | 67.80 | 70.10 | 79.15 | 66.23 | 82.17 | 62.24 |
| | pFedGraph | 63.46 | 85.33 | 51.19 | 0.0 | 57.30 | 67.28 | 62.11 | 78.89 | 63.75 | 82.13 | 61.14 |
| | FedAMP | 61.25 | 84.00 | 52.26 | 0.0 | 55.50 | 67.54 | 46.26 | 79.59 | 56.36 | 79.44 | 58.22 |

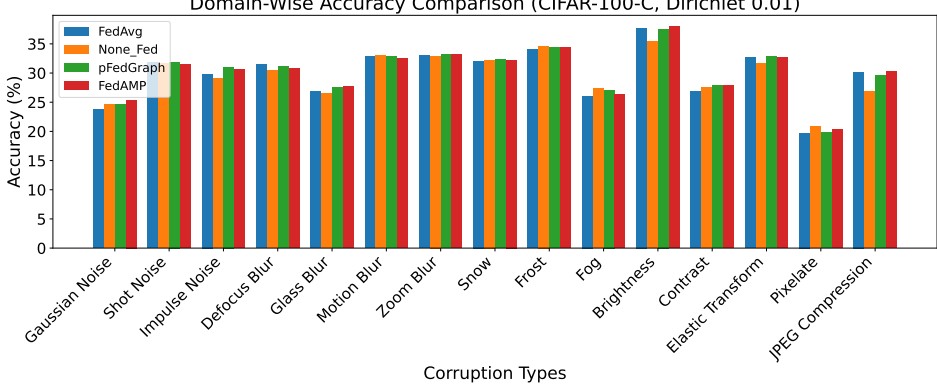

Figure 21: Domain-wise accuracy comparison of our method (Dirichlet $\delta = 0.01$) across CIFAR-100-C corruptions.

Table 4: Class-wise sample counts for CIFAR-10-C under Dirichlet heterogeneity $\delta = 0.01$, rounds 1–45. Each row shows how many examples of each class a client sees in that test round, illustrating extreme non-IID splits (e.g., class 3 never appears).

| Round | Class 0 | Class 1 | Class 2 | Class 3 | Class 4 | Class 5 | Class 6 | Class 7 | Class 8 | Class 9 |
|---|---|---|---|---|---|---|---|---|---|---|
| 1 | 496 | 5 | 268 | 0 | 200 | 0 | 327 | 200 | 304 | 200 |
| 2 | 266 | 0 | 199 | 0 | 202 | 132 | 400 | 200 | 401 | 200 |
| 3 | 200 | 0 | 0 | 0 | 200 | 200 | 400 | 200 | 400 | 400 |
| 4 | 200 | 0 | 0 | 0 | 200 | 115 | 400 | 200 | 485 | 400 |
| 5 | 200 | 0 | 0 | 0 | 53 | 10 | 400 | 200 | 736 | 401 |
| 6 | 496 | 5 | 268 | 0 | 200 | 0 | 327 | 200 | 304 | 200 |
| 7 | 266 | 0 | 199 | 0 | 202 | 132 | 400 | 200 | 401 | 200 |
| 8 | 200 | 0 | 0 | 0 | 200 | 200 | 400 | 200 | 400 | 400 |
| 9 | 200 | 0 | 0 | 0 | 200 | 115 | 400 | 200 | 485 | 400 |
| 10 | 200 | 0 | 0 | 0 | 53 | 10 | 400 | 200 | 736 | 401 |
| 11 | 496 | 5 | 268 | 0 | 200 | 0 | 327 | 200 | 304 | 200 |
| 12 | 266 | 0 | 199 | 0 | 202 | 132 | 400 | 200 | 401 | 200 |
| 13 | 200 | 0 | 0 | 0 | 200 | 200 | 400 | 200 | 400 | 400 |
| 14 | 200 | 0 | 0 | 0 | 200 | 115 | 400 | 200 | 485 | 400 |
| 15 | 200 | 0 | 0 | 0 | 53 | 10 | 400 | 200 | 736 | 401 |
| 16 | 496 | 5 | 268 | 0 | 200 | 0 | 327 | 200 | 304 | 200 |
| 17 | 266 | 0 | 199 | 0 | 202 | 132 | 400 | 200 | 401 | 200 |
| 18 | 200 | 0 | 0 | 0 | 200 | 200 | 400 | 200 | 400 | 400 |
| 19 | 200 | 0 | 0 | 0 | 200 | 115 | 400 | 200 | 485 | 400 |
| 20 | 200 | 0 | 0 | 0 | 53 | 10 | 400 | 200 | 736 | 401 |
| 21 | 496 | 5 | 268 | 0 | 200 | 0 | 327 | 200 | 304 | 200 |
| 22 | 266 | 0 | 199 | 0 | 202 | 132 | 400 | 200 | 401 | 200 |
| 23 | 200 | 0 | 0 | 0 | 200 | 200 | 400 | 200 | 400 | 400 |
| 24 | 200 | 0 | 0 | 0 | 200 | 115 | 400 | 200 | 485 | 400 |
| 25 | 200 | 0 | 0 | 0 | 53 | 10 | 400 | 200 | 736 | 401 |
| 26 | 496 | 5 | 268 | 0 | 200 | 0 | 327 | 200 | 304 | 200 |
| 27 | 266 | 0 | 199 | 0 | 202 | 132 | 400 | 200 | 401 | 200 |
| 28 | 200 | 0 | 0 | 0 | 200 | 200 | 400 | 200 | 400 | 400 |
| 29 | 200 | 0 | 0 | 0 | 200 | 115 | 400 | 200 | 485 | 400 |
| 30 | 200 | 0 | 0 | 0 | 53 | 10 | 400 | 200 | 736 | 401 |
| 31 | 496 | 5 | 268 | 0 | 200 | 0 | 327 | 200 | 304 | 200 |
| 32 | 266 | 0 | 199 | 0 | 202 | 132 | 400 | 200 | 401 | 200 |
| 33 | 200 | 0 | 0 | 0 | 200 | 200 | 400 | 200 | 400 | 400 |
| 34 | 200 | 0 | 0 | 0 | 200 | 115 | 400 | 200 | 485 | 400 |
| 35 | 200 | 0 | 0 | 0 | 53 | 10 | 400 | 200 | 736 | 401 |
| 36 | 496 | 5 | 268 | 0 | 200 | 0 | 327 | 200 | 304 | 200 |
| 37 | 266 | 0 | 199 | 0 | 202 | 132 | 400 | 200 | 401 | 200 |
| 38 | 200 | 0 | 0 | 0 | 200 | 200 | 400 | 200 | 400 | 400 |
| 39 | 200 | 0 | 0 | 0 | 200 | 115 | 400 | 200 | 485 | 400 |
| 40 | 200 | 0 | 0 | 0 | 53 | 10 | 400 | 200 | 736 | 401 |
| 41 | 496 | 5 | 268 | 0 | 200 | 0 | 327 | 200 | 304 | 200 |
| 42 | 266 | 0 | 199 | 0 | 202 | 132 | 400 | 200 | 401 | 200 |
| 43 | 200 | 0 | 0 | 0 | 200 | 200 | 400 | 200 | 400 | 400 |
| 44 | 200 | 0 | 0 | 0 | 200 | 115 | 400 | 200 | 485 | 400 |
| 45 | 200 | 0 | 0 | 0 | 53 | 10 | 400 | 200 | 736 | 401 |

Table 5: Class-wise sample counts for CIFAR-10-C under Dirichlet heterogeneity $\delta = 0.01$, rounds 46–75. Each row shows how many examples of each class a client sees in that test round, illustrating extreme non-IID splits (e.g., class 3 never appears).

| Round | Class 0 | Class 1 | Class 2 | Class 3 | Class 4 | Class 5 | Class 6 | Class 7 | Class 8 | Class 9 |
|-------|---------|---------|---------|---------|---------|---------|---------|---------|---------|---------|
| 46 | 496 | 5 | 268 | 0 | 200 | 0 | 327 | 200 | 304 | 200 |
| 47 | 266 | 0 | 199 | 0 | 202 | 132 | 400 | 200 | 401 | 200 |
| 48 | 200 | 0 | 0 | 0 | 200 | 200 | 400 | 200 | 400 | 400 |
| 49 | 200 | 0 | 0 | 0 | 200 | 115 | 400 | 200 | 485 | 400 |
| 50 | 200 | 0 | 0 | 0 | 53 | 10 | 400 | 200 | 736 | 401 |
| 51 | 496 | 5 | 268 | 0 | 200 | 0 | 327 | 200 | 304 | 200 |
| 52 | 266 | 0 | 199 | 0 | 202 | 132 | 400 | 200 | 401 | 200 |
| 53 | 200 | 0 | 0 | 0 | 200 | 200 | 400 | 200 | 400 | 400 |
| 54 | 200 | 0 | 0 | 0 | 200 | 115 | 400 | 200 | 485 | 400 |
| 55 | 200 | 0 | 0 | 0 | 53 | 10 | 400 | 200 | 736 | 401 |
| 56 | 496 | 5 | 268 | 0 | 200 | 0 | 327 | 200 | 304 | 200 |
| 57 | 266 | 0 | 199 | 0 | 202 | 132 | 400 | 200 | 401 | 200 |
| 58 | 200 | 0 | 0 | 0 | 200 | 200 | 400 | 200 | 400 | 400 |
| 59 | 200 | 0 | 0 | 0 | 200 | 115 | 400 | 200 | 485 | 400 |
| 60 | 200 | 0 | 0 | 0 | 53 | 10 | 400 | 200 | 736 | 401 |
| 61 | 496 | 5 | 268 | 0 | 200 | 0 | 327 | 200 | 304 | 200 |
| 62 | 266 | 0 | 199 | 0 | 202 | 132 | 400 | 200 | 401 | 200 |
| 63 | 200 | 0 | 0 | 0 | 200 | 200 | 400 | 200 | 400 | 400 |
| 64 | 200 | 0 | 0 | 0 | 200 | 115 | 400 | 200 | 485 | 400 |
| 65 | 200 | 0 | 0 | 0 | 53 | 10 | 400 | 200 | 736 | 401 |
| 66 | 496 | 5 | 268 | 0 | 200 | 0 | 327 | 200 | 304 | 200 |
| 67 | 266 | 0 | 199 | 0 | 202 | 132 | 400 | 200 | 401 | 200 |
| 68 | 200 | 0 | 0 | 0 | 200 | 200 | 400 | 200 | 400 | 400 |
| 69 | 200 | 0 | 0 | 0 | 200 | 115 | 400 | 200 | 485 | 400 |
| 70 | 200 | 0 | 0 | 0 | 53 | 10 | 400 | 200 | 736 | 401 |
| 71 | 496 | 5 | 268 | 0 | 200 | 0 | 327 | 200 | 304 | 200 |
| 72 | 266 | 0 | 199 | 0 | 202 | 132 | 400 | 200 | 401 | 200 |
| 73 | 200 | 0 | 0 | 0 | 200 | 200 | 400 | 200 | 400 | 400 |
| 74 | 200 | 0 | 0 | 0 | 200 | 115 | 400 | 200 | 485 | 400 |
| 75 | 200 | 0 | 0 | 0 | 53 | 10 | 400 | 200 | 736 | 401 |

Table 6: Domain-wise average accuracy (%) on CIFAR-10-C (severity-5) under Dirichlet heterogeneity $\delta = 0.01$, comparing five TTA methods (TENT, CoTTA, ROID, RoTTA, BBN) across four federation schemes: FedAvg, None-Fed, pFedGraph, and FedAMP. Each block of five columns corresponds to one federation strategy, with "None-Fed" values indicating no federated strategy. The accuracy values under 15 corruption types are shown in the table.

| Corruption | FedAvg | | | | | None-Fed | | | | |
|---|---|---|---|---|---|---|---|---|---|---|
| | Tent | CoTTA | ROID | RoTTA | BBN | Tent | CoTTA | ROID | RoTTA | BBN |
| Gaussian Noise | 27.61 | 38.48 | 29.54 | 49.35 | 50.81 | 27.60 | 32.90 | 29.53 | 38.21 | 48.39 |
| Shot Noise | 27.93 | 43.30 | 29.57 | 58.66 | 64.45 | 27.86 | 36.98 | 29.60 | 32.25 | 56.37 |
| Impulse Noise | 25.49 | 36.81 | 26.60 | 48.14 | 51.47 | 25.46 | 33.53 | 26.61 | 41.64 | 37.20 |
| Defocus Blur | 30.88 | 40.44 | 33.11 | 49.99 | 50.70 | 30.82 | 40.44 | 33.08 | 14.10 | 38.01 |
| Glass Blur | 24.02 | 39.19 | 25.10 | 54.35 | 56.74 | 24.01 | 30.82 | 25.16 | 37.63 | 36.58 |
| Motion Blur | 30.92 | 46.89 | 32.67 | 62.86 | 66.80 | 30.92 | 40.37 | 32.69 | 33.85 | 51.72 |
| Zoom Blur | 30.76 | 46.89 | 33.04 | 63.01 | 70.50 | 30.68 | 46.89 | 33.02 | 65.85 | 59.04 |
| Snow | 29.75 | 50.29 | 31.35 | 72.82 | 68.50 | 29.76 | 51.16 | 31.36 | 48.01 | 58.49 |
| Frost | 30.55 | 53.32 | 32.10 | 76.09 | 76.74 | 30.54 | 47.54 | 32.08 | 50.95 | 68.36 |
| Fog | 31.30 | 49.44 | 33.01 | 67.57 | 73.26 | 31.29 | 49.44 | 32.98 | 52.19 | 60.41 |
| Brightness | 32.54 | 56.46 | 34.25 | 81.75 | 81.17 | 32.43 | 56.03 | 34.28 | 46.79 | 74.21 |
| Contrast | 31.18 | 39.17 | 32.82 | 47.16 | 49.33 | 31.10 | 36.03 | 32.83 | 9.07 | 41.22 |
| Elastic Transform | 28.05 | 42.33 | 28.76 | 56.61 | 61.05 | 28.05 | 36.57 | 28.76 | 17.90 | 52.74 |
| Pixelate | 28.97 | 36.65 | 30.22 | 44.33 | 50.08 | 28.97 | 40.20 | 30.19 | 33.02 | 47.09 |
| JPEG Compression | 26.89 | 39.17 | 27.45 | 57.82 | 62.05 | 26.90 | 33.90 | 27.46 | 24.82 | 56.64 |

| Corruption | pFedGraph | | | | | FedAMP | | | | |
|---|---|---|---|---|---|---|---|---|---|---|
| | Tent | CoTTA | ROID | RoTTA | BBN | Tent | CoTTA | ROID | RoTTA | BBN |
| Gaussian Noise | 50.81 | 39.21 | 29.52 | 50.81 | 52.58 | 50.75 | 39.21 | 29.48 | 50.75 | 52.35 |
| Shot Noise | 59.36 | 43.64 | 29.59 | 59.36 | 64.08 | 60.88 | 44.41 | 29.54 | 60.88 | 61.31 |
| Impulse Noise | 49.04 | 37.27 | 26.62 | 49.04 | 52.84 | 50.59 | 38.05 | 26.49 | 50.59 | 51.98 |
| Defocus Blur | 49.97 | 40.42 | 33.11 | 49.97 | 47.74 | 44.32 | 37.63 | 32.92 | 44.32 | 44.75 |
| Glass Blur | 54.27 | 39.15 | 25.10 | 54.27 | 54.30 | 54.23 | 39.12 | 24.96 | 54.23 | 48.67 |
| Motion Blur | 62.99 | 46.95 | 32.74 | 62.99 | 64.46 | 60.95 | 45.95 | 32.60 | 60.95 | 58.09 |
| Zoom Blur | 62.88 | 46.92 | 33.04 | 62.88 | 68.93 | 63.27 | 46.86 | 32.95 | 63.27 | 66.86 |
| Snow | 72.95 | 50.36 | 31.37 | 72.95 | 66.66 | 70.67 | 48.05 | 31.19 | 70.67 | 64.76 |
| Frost | 76.10 | 53.35 | 32.05 | 76.10 | 76.49 | 74.63 | 52.04 | 31.92 | 74.63 | 73.62 |
| Fog | 67.55 | 49.43 | 33.00 | 67.55 | 70.61 | 64.76 | 44.66 | 32.88 | 64.76 | 68.73 |
| Brightness | 81.77 | 56.49 | 34.21 | 81.77 | 80.76 | 81.02 | 53.95 | 34.10 | 81.02 | 77.39 |
| Contrast | 47.40 | 39.22 | 32.86 | 47.40 | 47.40 | 39.90 | 39.21 | 32.72 | 39.90 | 45.21 |
| Elastic Transform | 56.39 | 35.52 | 28.75 | 56.39 | 59.69 | 55.67 | 36.88 | 28.58 | 55.67 | 53.76 |
| Pixelate | 42.07 | 42.61 | 30.17 | 42.07 | 49.52 | 44.79 | 42.74 | 30.14 | 44.79 | 48.36 |
| JPEG Compression | 58.32 | 39.29 | 27.48 | 58.32 | 61.09 | 58.56 | 42.74 | 27.38 | 58.56 | 57.48 |

Table 7: Client-wise accuracy (%) on CIFAR-10-C under Dirichlet $\delta = 0.01$ Accuracy is reported for each client (0–9) using five TTA methods (Tent, CoTTA, ROID, RoTTA, BBN) across four federated learning setups: FedAvg, pFedGraph, FedAMP, and None-Fed. The bottom row shows the average accuracy across clients, highlighting the impact of personalization under non-IID settings.

| Client | FedAvg | | | | | None-Fed | | | | |
|---|---|---|---|---|---|---|---|---|---|---|
| | Tent | CoTTA | ROID | RoTTA | BBN | Tent | CoTTA | ROID | RoTTA | BBN |
| 0 | 18.40 | 22.74 | 18.79 | 27.08 | 28.08 | 18.06 | 17.57 | 18.95 | 16.66 | 17.66 |
| 1 | 2.02 | 4.32 | 2.02 | 6.62 | 7.62 | 2.02 | 4.32 | 2.03 | 4.07 | 5.07 |
| 2 | 2.03 | 4.55 | 2.10 | 7.07 | 8.07 | 2.02 | 4.55 | 2.13 | 4.91 | 5.91 |
| 3 | 13.19 | 16.58 | 13.82 | 19.97 | 20.97 | 13.13 | 16.05 | 13.83 | 10.77 | 11.77 |
| 4 | 5.41 | 8.88 | 5.62 | 12.36 | 13.36 | 5.38 | 8.88 | 5.72 | 8.42 | 9.42 |
| 5 | 17.28 | 19.11 | 17.69 | 20.94 | 21.94 | 17.33 | 19.11 | 17.72 | 13.86 | 14.86 |
| 6 | 13.57 | 15.75 | 14.06 | 16.43 | 17.43 | 13.53 | 15.75 | 14.02 | 11.66 | 12.66 |
| 7 | 2.02 | 4.13 | 2.31 | 6.23 | 7.23 | 2.00 | 4.13 | 2.32 | 4.16 | 5.16 |
| 8 | 10.41 | 12.31 | 10.46 | 14.16 | 15.16 | 10.41 | 12.31 | 11.18 | 7.77 | 8.77 |
| 9 | 2.29 | 4.46 | 2.59 | 6.63 | 7.63 | 2.27 | 4.46 | 2.56 | 3.32 | 4.32 |
| **Mean** | 8.66 | 11.21 | 8.95 | 13.75 | 14.75 | 8.61 | 11.21 | 9.05 | 8.56 | 9.56 |

| Client | pFedGraph | | | | | FedAMP | | | | |
|---|---|---|---|---|---|---|---|---|---|---|
| | Tent | CoTTA | ROID | RoTTA | BBN | Tent | CoTTA | ROID | RoTTA | BBN |
| 0 | 18.40 | 22.74 | 18.79 | 27.45 | 28.45 | 18.40 | 22.74 | 18.76 | 27.09 | 28.09 |
| 1 | 2.02 | 4.32 | 2.02 | 6.38 | 7.38 | 2.02 | 4.32 | 2.00 | 6.15 | 7.15 |
| 2 | 2.03 | 4.55 | 2.10 | 7.14 | 8.14 | 2.03 | 4.55 | 2.08 | 6.78 | 7.78 |
| 3 | 13.19 | 16.58 | 13.83 | 20.10 | 21.10 | 13.19 | 16.58 | 13.81 | 20.14 | 21.14 |
| 4 | 5.41 | 8.88 | 5.61 | 12.44 | 13.44 | 5.41 | 8.88 | 5.59 | 11.65 | 12.65 |
| 5 | 17.28 | 19.11 | 17.69 | 20.56 | 21.56 | 17.29 | 19.11 | 17.69 | 20.30 | 21.30 |
| 6 | 13.57 | 15.75 | 14.06 | 16.97 | 17.97 | 13.57 | 15.75 | 14.04 | 16.31 | 17.31 |
| 7 | 2.02 | 4.13 | 2.31 | 6.28 | 7.28 | 2.03 | 4.13 | 2.29 | 6.38 | 7.38 |
| 8 | 10.41 | 12.31 | 10.45 | 14.27 | 15.27 | 10.41 | 12.31 | 10.44 | 14.16 | 15.16 |
| 9 | 2.29 | 4.46 | 2.60 | 6.73 | 7.73 | 2.29 | 4.46 | 2.59 | 6.72 | 7.72 |
| **Mean** | 8.66 | 11.21 | 8.95 | 13.83 | 14.83 | 8.66 | 11.21 | 8.93 | 13.57 | 14.57 |