# OpenReview forum: "pFedBBN: A Personalized Federated Test-Time Adaptation with Balanced Batch Normalization for Class-Imbalanced Data"
_ICLR.cc/2026/Conference — Submitted to ICLR 2026_

### Official Review · Reviewer_mNWi · 2025-10-26

**Soundness:** 2
**Presentation:** 3
**Contribution:** 2
**Rating:** 2
**Confidence:** 5

**Summary:**

This paper proposes pFedBBN, a personalized federated testing-time adaptation (FTTA) framework that combines Balanced Batch Normalization (BBN) and class-wise adaptive normalization (CWAN) to mitigate the negative effects of class imbalance under distribution shifts. It further introduces a BBN-statistics-based client similarity to guide personalized aggregation, with the goal of improving robustness and minority-class performance in federated environments.

**Strengths:**

1. The work tackles the intersection of test-time adaptation + class imbalance + personalization, a topic of growing interest.

2. Using class-wise BBN statistics estimated via pseudo labels, and leveraging them for similarity-aware aggregation, is conceptually clear.

**Weaknesses:**

1. Potential contradiction in stating “only affine parameters are updated” while simultaneously updating and using class-wise BBN running statistics. The interaction between the two must be clarified.

2. Dividing by feature dimension d in Eq.(2)/(3) is atypical for BN statistics. The promised “interpolation with batch statistics” in contributions is not visible in Eq.(4), which is a uniform mean over classes.

3. Eq.(7) averages raw L2 distances across layers without normalization, which may be unstable across architectures. No sensitivity analysis provided.

4. In Table 1, BBN alone sometimes outperforms pFedBBN (e.g., CIFAR-10-C IID: 72.61 vs. 70.11), suggesting personalized aggregation can hurt when data are IID. Assertions should be toned down.

5. Claims about improving minority-class performance require macro-F1 / balanced accuracy / per-class recall in the main table, currently insufficient evidence.

6. FedCTTA (2025) and other recent FTTA methods are mentioned but not compared.

7. The paper claims class-balanced BN statistics and similarity-based aggregation drive the gains, but lacks direct interpretability to verify this (e.g., similarity matrix visualization, minority-class metrics, sensitivity analysis).

8. Experiments are conducted only on small-scale datasets, lacking validation on larger and more realistic federated benchmarks. This raises concerns about the reliability and scalability of the reported performance.

**Questions:**

1. Are BBN running statistics also updated during TTA, or are they frozen while only γ/β are optimized?

2. How is the “interpolation with batch statistics” implemented? Why does Eq.(4) show uniform class averaging?

3. What is the granularity/frequency of sharing BN stats (per-channel? per-layer? every round?) and the bandwidth?

4. Why does pFedBBN sometimes underperform local BBN in IID settings?

5. What concrete privacy guarantees are assumed when sharing high-dimensional BN statistics?

---

### Official Review · Reviewer_uDPV · 2025-10-27

**Soundness:** 2
**Presentation:** 1
**Contribution:** 2
**Rating:** 2
**Confidence:** 3

**Summary:**

This paper proposes pFedBBN, a personalized federated test-time adaptation framework that aims to handle class imbalance and domain shifts in federated learning without labeled data.

**Strengths:**

The studied topic is interesting and important. The experiment results seem to be encouraging.

**Weaknesses:**

1. The literature review is not comprehensive. The data heterogeneity, particularly as a form of data imbalance and/or long-tail, has been extensively studied by the community, e.g., [1] and [2].


2. The presentation of the methodology section is very sloppy making it very hard to accurately understand the details. The experiemnt sections failed to disclose necessary details. Therefore, it's hard to assess the contribution and novelty. Please see the `Question` sections for details.

[1] Shang, Xinyi, et al. "Federated Learning on Heterogeneous and Long-Tailed Data via Classifier Re-Training with Federated Features."

[2] Dai, Yutong, Zeyuan Chen, Junnan Li, Shelby Heinecke, Lichao Sun, and Ran Xu. "Tackling data heterogeneity in federated learning with class prototypes." In Proceedings of the AAAI Conference on Artificial Intelligence, vol. 37, no. 6, pp. 7314-7322. 2023.

**Questions:**

1. What is Balanced Batch Normalization (BBN)? Is it the same as proposed CWAN? Some parts of the paper mention "each client performs unsupervised local adaptation using class-wise balanced batch normalization (BBN)" and some parts mention "Prior to adaptation, the student model is equipped with CWAN by replacing standard batch normalization layers with class-aware normalization layers, enabling balanced statistics across classes during inference".

2. How is the pseudo-label $\hat c_i$ computed (section 3.1.1)? Is it computed from the pre-trained source model? What is the difference between the source model and the teacher model? What is the augmented counterpart $\tilde x$ defined in section 3.1.2?

3. CWAN employs momentum-based running updates (Eqs. 1–3).
    * What is the rationale for using running batch updates instead of computing statistics directly using the client's data? I guess these are intended to be used in the local client adaptation, i.e., in each step of updating BBN or CWAN paramters. In other words,  But I'm not sure since Section 3.1 is poorly presented and I can only make some guess.
    * The updating rule  Eq. (2) needs more explanations. It seems to be related to Welford online variance update formula.
    * How are absent classes handled in Eqs. (1–4)? Are their statistics frozen, interpolated, or reinitialized?
    * How sensitive is CWAN to pseudo-label noise, especially under heavy class imbalance?
    * Both Eq. (2) and Eq. (3) divide the update terms by the feature dimension $d$, which is nonstandard in batch normalization. What is the theoretical or empirical motivation for dividing by $d$?

4. What is the final loss function used for local client adaptation (section 3.1.2)? It only introduced $L_{KD}$ and  $L_{CR}$.

5. The matrix $W$ in Eq. (8)is described as row-stochastic, and later described as symmetric distance matrix in the Section 4.2.  row-stochastic and symmetric are not equivalent.

6. Does the model aggregation happen on the server side or the client side?

7. It's unclear how the hyperparameters are chosen and how they impact the results, e.g., $\eta$ in Eq. (3), $\delta$ in Eq. (5) (which conflicts the parameter used for the Dirichlet distribution), and $\tau$ used in Eq. (8). It's unclear how the source model and teacher models are obtained / constructed for each client.


**Minor Comments**
1. Failed to disclose the usage of large language model as required.
2. Please use  \citep and \citet Tex commands properly.

---

### Official Review · Reviewer_Wmpd · 2025-10-27

**Soundness:** 2
**Presentation:** 3
**Contribution:** 2
**Rating:** 6
**Confidence:** 4

**Summary:**

This paper introduces pFedBBN, a novel personalized federated test-time adaptation (FTTA) framework designed to address the combined challenges of domain shifts and severe class imbalance (CI). The core issue identified is that standard TTA methods fail in federated, non-IID settings because conventional batch normalization statistics become biased by dominant classes, degrading performance on rare classes. pFedBBN tackles this via a two-part, privacy-preserving approach: first, clients perform fully unsupervised local adaptation using a Class-Wise Adaptive Normalization (CWAN) module. This module employs Balanced Batch Normalization (BBN) to track per-class feature statistics using pseudo-labels, thereby mitigating majority-class bias. This local update is stabilized using a confidence-filtered knowledge distillation process to reduce noise from pseudo-labels. Second, clients exchange only their BBN statistics, which act as privacy-preserving domain descriptors. A server then computes an inter-client similarity matrix from these statistics to perform a personalized, weighted aggregation, ensuring clients collaborate primarily with domain-similar peers.

**Strengths:**

1. First to address the combination of Class Imbalance and Federated Test-time Adaptation
2. Part of the proposed method, the balanced batch normalization, can be integrated with prior works for improved performance.
3. Experiment shows improved results compared to baseline methods.

**Weaknesses:**

1. The work lacks an explanation of the concrete design of the baseline methods; thus, it is not clear how the proposed method is novel compared to prior methods.
2. The idea of managing class-wise statistics and dealing with batch normalization is similar to several more recent related works dealing with class imbalance problems [1,2,3,4,5], but the relevant works are not cited or discussed.
3. The algorithm pseudo-code is not provided.
4. The experiment is conducted on limited datasets of CIFAR 10 and 100, but is not conducted on the more challenging Digits-5 and PACS

[1] Rebalancing batch normalization for exemplar-based class-incremental learning
[2] FedBN: Federated Learning on Non-IID Features via Local Batch Normalization.
[3] FedGCR: Achieving Performance and Fairness for Federated Learning with Distinct Client Types via Group Customization and Reweighting
[4] TTA-FedDG: Leveraging Test-Time Adaptation to Address Federated Domain Generalization
[5] Latte: Collaborative Test-Time Adaptation of Vision-Language Models in Federated Learning

Other issues:
- The citation format is incorrect throughout the paper

**Questions:**

1. What is the novel contribution of the proposed method compared to more recent class imbalance works?
2. Can you provide the concrete pseudo-code of your proposed method?
3. Does the proposed method work on class imbalance settings with more distinct classes?

---

### Official Review · Reviewer_5Gvb · 2025-11-05

**Soundness:** 2
**Presentation:** 2
**Contribution:** 2
**Rating:** 2
**Confidence:** 4

**Summary:**

The paper “pFedBBN: Personalized Federated Test-Time Adaptation with Balanced Batch Normalization for Class-Imbalanced Data” addresses the challenge of adapting federated models to unseen, unlabeled, and class-imbalanced domains. Standard BN layers bias toward head classes, harming tail performance. The authors propose pFedBBN, which combines Balanced Batch Normalization (BBN)—maintaining class-wise BN statistics via pseudo-labels—with a teacher–student distillation for unsupervised local adaptation. Only BN affine parameters are updated for efficiency. Clients then exchange BN statistics to compute domain similarity and perform personalized, similarity-weighted aggregation instead of uniform averaging. Experiments on CIFAR-10/100-C show pFedBBN outperforms prior FL and TTA methods, especially for minority classes, offering a lightweight, privacy-preserving solution to non-IID and imbalanced federated test-time scenarios.

**Strengths:**

Originality:
The paper introduces a novel perspective on federated test-time adaptation (FTTA) by explicitly tackling class imbalance, a rarely addressed challenge in this context. The proposed Balanced Batch Normalization (BBN) and class-wise adaptive normalization represent an original and conceptually elegant modification to standard BN, ensuring fair adaptation across majority and minority classes. Integrating this with a similarity-based personalized aggregation using BN statistics is a creative and privacy-preserving innovation that bridges personalization and adaptation in FL.

Quality:
The methodology is well-grounded, combining sound theoretical intuition (balanced statistics to mitigate bias) with a practical, lightweight implementation (updating only BN affine parameters). The paper offers strong empirical validation across multiple datasets and imbalance levels, showing consistent gains over state-of-the-art FTTA and FL baselines. The use of pseudo-label filtering and teacher–student regularization further strengthens the robustness of the approach.

Clarity:
The paper is clearly written and logically structured, with each component—local adaptation, normalization, and aggregation—introduced and motivated in a coherent flow. Equations are well-formulated, and visual illustrations effectively convey the model pipeline. The balance between technical depth and readability is maintained throughout.

Significance:
This work makes an important step toward practical deployment of FL models under realistic conditions—where class imbalance, domain shift, and unlabeled test data coexist. By reducing head-class bias and enabling personalized test-time adaptation, pFedBBN advances the frontier of robust, generalizable, and privacy-aware federated learning. Its modular design also makes it extendable to broader multimodal or resource-constrained FL applications.

**Weaknesses:**

Limited empirical scope:
The experiments are confined to CIFAR-10-C and CIFAR-100-C—synthetic corruption benchmarks that only approximate domain shifts. This leaves uncertainty about the framework’s robustness in real-world federated settings (e.g., medical imaging, speech, or sensor data). Evaluations on larger or more heterogeneous datasets would better validate generality.

Dependence on pseudo-labels:
The class-wise BN statistics rely on pseudo-labels generated by the teacher. Under severe class imbalance, tail classes may receive few confident samples, leading to inaccurate statistics and residual bias. The paper could strengthen its approach by proposing confidence calibration or reweighted statistic estimation for low-frequency classes.

Limited adaptation capacity:
Only BN affine parameters are updated during test-time adaptation. While efficient, this design may be insufficient for large domain shifts. Incorporating adapter modules or low-rank parameter tuning (e.g., LoRA) could enhance flexibility without heavy computation.

Similarity metric simplicity:
The inter-client similarity uses Euclidean distance on flattened BN statistics, which might not fully capture semantic or feature-level domain divergence. A learned or information-theoretic similarity metric could provide more meaningful personalization.

Ablation and sensitivity gaps:
The paper would benefit from ablation studies isolating the effects of BBN, pseudo-label confidence thresholding, and the aggregation strategy. Sensitivity analyses on the temperature parameter
𝜏
τ in similarity weighting are also missing.

Theoretical justification:
While empirically motivated, the paper lacks formal analysis connecting class-wise normalization to reduced bias or improved convergence in federated adaptation. Including theoretical insights or convergence bounds would reinforce the method’s credibility.

**Questions:**

How robust is pFedBBN to noisy pseudo-labels, especially for tail classes—did you explore adaptive confidence thresholds or uncertainty calibration? Updating only BN affine parameters may limit adaptation; could lightweight adapters (e.g., LoRA) improve flexibility? Why use Euclidean distance for similarity—have you compared cosine or learned metrics? Does personalized aggregation risk bias toward large or clean clients? Please provide ablations for each module and sensitivity to τ. Have you tested on real-world non-IID datasets beyond CIFAR-C? Finally, any theoretical or interpretive explanation for how balanced BN reduces head-class bias would strengthen the work.

---

### Meta-Review · Area_Chair_xWeD · 2026-01-07

**Summary:**

Reviewer Wmpd has summarized this paper particularly well so I quoted the reviewer's summary here:

This paper introduces pFedBBN, a novel personalized federated test-time adaptation (FTTA) framework designed to address the combined challenges of domain shifts and severe class imbalance (CI). The core issue identified is that standard TTA methods fail in federated, non-IID settings because conventional batch normalization statistics become biased by dominant classes, degrading performance on rare classes. pFedBBN tackles this via a two-part, privacy-preserving approach: first, clients perform fully unsupervised local adaptation using a Class-Wise Adaptive Normalization (CWAN) module. This module employs Balanced Batch Normalization (BBN) to track per-class feature statistics using pseudo-labels, thereby mitigating majority-class bias. This local update is stabilized using a confidence-filtered knowledge distillation process to reduce noise from pseudo-labels. Second, clients exchange only their BBN statistics, which act as privacy-preserving domain descriptors. A server then computes an inter-client similarity matrix from these statistics to perform a personalized, weighted aggregation, ensuring clients collaborate primarily with domain-similar peers.

--

Overall, the idea is fairly interesting but there are multiple critical concerns on lacking theoretical justification, limited empirical scopes, lacking ablation studies, and more. The original scores are overwhelmingly leaning towards rejection. As the authors have not provided a rebuttal, I don't think the rejection sentiment will change.

**Reviewer Concerns:**

There is no rebuttal.

**Reviewer Scores:**

There is no rebuttal so the scores will not change.

---

### Decision · Program_Chairs · 2026-01-26

Reject